# Fantope Projection and Selection:
# A near-optimal convex relaxation of sparse PCA

**Vincent Q. Vu**
The Ohio State University
vqv@stat.osu.edu

**Juhee Cho**
University of Wisconsin, Madison
chojuhee@stat.wisc.edu

**Jing Lei**
Carnegie Mellon University
leij09@gmail.com

**Karl Rohe**
University of Wisconsin, Madison
karlrohe@stat.wisc.edu

## Abstract

We propose a novel convex relaxation of sparse principal subspace estimation based on the convex hull of rank-$d$ projection matrices (the *Fantope*). The convex problem can be solved efficiently using alternating direction method of multipliers (ADMM). We establish a near-optimal convergence rate, in terms of the sparsity, ambient dimension, and sample size, for estimation of the principal subspace of a general covariance matrix without assuming the spiked covariance model. In the special case of $d = 1$, our result implies the near-optimality of DSPCA (d'Aspremont et al. [1]) even when the solution is not rank 1. We also provide a general theoretical framework for analyzing the statistical properties of the method for arbitrary input matrices that extends the applicability and provable guarantees to a wide array of settings. We demonstrate this with an application to Kendall's tau correlation matrices and transelliptical component analysis.

## 1   Introduction

Principal components analysis (PCA) is a popular technique for unsupervised dimension reduction that has a wide range of application—science, engineering, and any place where multivariate data is abundant. PCA uses the eigenvectors of the sample covariance matrix to compute the linear combinations of variables with the largest variance. These principal directions of variation explain the covariation of the variables and can be exploited for dimension reduction. In contemporary applications where variables are plentiful (large $p$) but samples are relatively scarce (small $n$), PCA suffers from two major weaknesses : 1) the interpretability and subsequent use of the principal directions is hindered by their dependence on *all* of the variables; 2) it is generally inconsistent in high-dimensions, i.e. the estimated principal directions can be noisy and unreliable [see 2, and the references therein].

Over the past decade, there has been a fever of activity to address the drawbacks of PCA with a class of techniques called *sparse PCA* that combine the essence of PCA with the assumption that the phenomena of interest depend mostly on a few variables. Examples include algorithmic [e.g., 1, 3–10] and theoretical [e.g., 11–14] developments. However, much of this work has focused on the first principal component. One rationale behind this focus is by analogy with ordinary PCA: additional components can be found by iteratively deflating the input matrix to account for variation uncovered by previous components. However, the use of deflation with sparse PCA entails complications of non-orthogonality, sub-optimality, and multiple tuning parameters [15]. Identifiability and consistency present more subtle issues. The principal directions of variation correspond to eigenvectors of some population matrix $\Sigma$. There is no reason to assume a priori that the $d$ largest eigenvalues

of $\Sigma$ are distinct. Even if the eigenvalues are distinct, estimates of individual eigenvectors can be unreliable if the gap between their eigenvalues is small. So it seems reasonable, if not necessary, to de-emphasize eigenvectors and to instead focus on their span, i.e. the *principal subspace* of variation.

There has been relatively little work on the problem of estimating the principal subspace or even multiple eigenvectors simultaneously. Most works that do are limited to iterative deflation schemes or optimization problems whose global solution is intractable to compute. Sole exceptions are the *diagonal thresholding* method [2], which is just ordinary PCA applied to the subset of variables with largest marginal sample variance, or refinements such as iterative thresholding [16], which use diagonal thresholding as an initial estimate. These works are limited, because they cannot be used when the variables have equal variances (e.g., correlation matrices). Theoretical results are equally limited in their applicability. Although the optimal minimax rates for the sparse principal subspace problem are known in both the spiked [17] and general [18] covariance models, existing statistical guarantees only hold under the restrictive *spiked covariance model*, which essentially guarantees that diagonal thresholding has good properties, or for estimators that are computationally intractable.

In this paper, we propose a novel convex optimization problem to estimate the $d$-dimensional principal subspace of a population matrix $\Sigma$ based on a noisy input matrix $S$. We show that if $S$ is a sample covariance matrix and the projection $\Pi$ of the $d$-dimensional principal subspace of $\Sigma$ depends only on $s$ variables, then with a suitable choice of regularization parameter, the Frobenius norm of the error of our estimator $\widehat{X}$ is bounded with high probability

$$\|\|\widehat{X} - \Pi\|\|_2 = O\big((\lambda_1/\delta)s\sqrt{\log p/n}\big)$$

where $\lambda_1$ is the largest eigenvalue of $\Sigma$ and $\delta$ the gap between the $d$th and $(d+1)$th largest eigenvalues of $\Sigma$. This rate turns out to be nearly minimax optimal (Corollary 3.3), and under additional assumptions on signal strength, it also allows us to recover the support of the principal subspace (Theorem 3.2). Moreover, we provide easy to verify conditions (Theorem 3.3) that yield near-optimal statistical guarantees for other choices of input matrix, such as Pearson's correlation and Kendall's tau correlation matrices (Corollary 3.4).

Our estimator turns out to be a semidefinite program (SDP) that generalizes the DSPCA approach of [1] to $d \geq 1$ dimensions. It is based on a convex body, called the *Fantope*, that provides a tight relaxation for simultaneous rank and orthogonality constraints on the positive semidefinite cone. Solving the SDP is non-trivial. We find that an alternating direction method of multipliers (ADMM) algorithm [e.g., 19] can efficiently compute its global optimum (Section 4).

In summary, the main contributions of this paper are as follows.

1. We formulate the sparse principal subspace problem as a novel semidefinite program with a Fantope constraint (Section 2).

2. We show that the proposed estimator achieves a near optimal rate of convergence in subspace estimation without assumptions on the rank of the solution or restrictive spiked covariance models. This is a first for both $d = 1$ and $d > 1$ (Section 3).

3. We provide a general theoretical framework that accommodates other matrices, in addition to sample covariance, such as Pearson's correlation and Kendall's tau.

4. We develop an efficient ADMM algorithm to solve the SDP (Section 4), and provide numerical examples that demonstrate the superiority of our approach over deflation methods in both computational and statistical efficiency (Section 5).

The remainder of the paper explains each of these contributions in detail, but we defer all proofs to Appendix A.

**Related work**  Existing work most closely related to ours is the DSPCA approach for single component sparse PCA that was first proposed in [1]. Subsequently, there has been theoretical analysis under a spiked covariance model and restrictions on the entries of the eigenvectors [11], and algorithmic developments including block coordinate ascent [9] and ADMM [20]. The crucial difference with our work is that this previous work only considered $d = 1$. The $d > 1$ case requires invention and novel techniques to deal with a convex body, the Fantope, that has never before been used in sparse PCA.

**Notation** For matrices $A, B$ of compatible dimension $\langle A, B \rangle := \operatorname{tr}(A^T B)$ is the Frobenius inner product, and $\|A\|_2^2 := \langle A, A \rangle$ is the squared Frobenius norm. $\|x\|_q$ is the usual $\ell_q$ norm with $\|x\|_0$ defined as the number of nonzero entries of $x$. $\|A\|_{a,b}$ is the $(a, b)$-norm defined to be the $\ell_b$ norm of the vector of rowwise $\ell_a$ norms of $A$, e.g. $\|A\|_{1,\infty}$ is the maximum absolute row sum. For a symmetric matrix $A$, we define $\lambda_1(A) \geq \lambda_2(A) \geq \cdots$ to be the eigenvalues of $A$ with multiplicity. When the context is obvious we write $\lambda_j := \lambda_j(A)$ as shorthand. For two subspaces $\mathcal{M}_1$ and $\mathcal{M}_2$, $\sin \Theta(\mathcal{M}_1, \mathcal{M}_2)$ is defined to be the matrix whose diagonals are the sines of the canonical angles between the two subspaces [see 21, §VII].

## 2 Sparse subspace estimation

Given a symmetric input matrix $S$, we propose a sparse principal subspace estimator $\widehat{X}$ that is defined to be a solution of the semidefinite program

$$
\begin{aligned}
\text{maximize} \quad & \langle S, X \rangle - \lambda \|X\|_{1,1} \\
\text{subject to} \quad & X \in \mathcal{F}^d,
\end{aligned}
\tag{1}
$$

in the variable $X$, where

$$
\mathcal{F}^d := \left\{ X : 0 \preceq X \preceq I \text{ and } \operatorname{tr}(X) = d \right\}
$$

is a convex body called the **Fantope** [22, §2.3.2], and $\lambda \geq 0$ is a regularization parameter that encourages sparsity. When $d = 1$, the spectral norm bound in $\mathcal{F}^d$ is redundant and (1) reduces to the DSPCA approach of [1]. The motivation behind (1) is based on two key insights.

The first insight is a variational characterization of the principal subspace of a symmetric matrix. The sum of the $d$ largest eigenvalues of a symmetric matrix $A$ can be expressed as

$$
\sum_{i=1}^d \lambda_i(A) \overset{(a)}{=} \max_{V^T V = I_d} \langle A, VV^T \rangle \overset{(b)}{=} \max_{X \in \mathcal{F}^d} \langle A, X \rangle .
\tag{2}
$$

Identity (a) is an extremal property known as *Ky Fan's maximum principle* [23]; (b) is based on the less well known observation that

$$
\mathcal{F}^d = \operatorname{conv}(\{VV^T : V^T V = I_d\}) ,
$$

i.e. the extremal points of $\mathcal{F}^d$ are the rank-$d$ projection matrices. See [24] for proofs of both.

The second insight is a connection between the $(1, 1)$-norm and a notion of subspace sparsity introduced by [18]. Any $X \succeq 0$ can be factorized (non-uniquely) as $X = VV^T$.

**Lemma 2.1.** *If $X = VV^T$, then $\|X\|_{1,1} \leq \|V\|_{2,1}^2 \leq \|V\|_{2,0}^2 \operatorname{tr}(X)$.*

Consequently, any $X \in \mathcal{F}^d$ that has at most $s$ non-zero rows necessarily has $\|X\|_{1,1} \leq s^2 d$. Thus, $\|X\|_{1,1}$ is a convex relaxation of what [18] call *row sparsity* for subspaces.

These two insights reveal that (1) is a semidefinite relaxation of the non-convex problem

$$
\begin{aligned}
\text{maximize} \quad & \langle S, VV^T \rangle - \lambda \|V\|_{2,0}^2 d \\
\text{subject to} \quad & V^T V = I_d .
\end{aligned}
$$

[18] proposed solving an equivalent form of the above optimization problem and showed that the estimator corresponding to its global solution is minimax rate optimal under a general statistical model for $S$. Their estimator requires solving an NP-hard problem. The advantage of (1) is that it is *computationally tractable*.

**Subspace estimation** The constraint $\widehat{X} \in \mathcal{F}^d$ guarantees that its rank is $\geq d$. However $\widehat{X}$ need not be an extremal point of $\mathcal{F}^d$, i.e. a rank-$d$ projection matrix. In order to obtain a proper $d$-dimensional subspace estimate, we can extract the $d$ leading eigenvectors of $\widehat{X}$, say $\widehat{V}$, and form the projection matrix $\widehat{\Pi} = \widehat{V}\widehat{V}^T$. The projection is unique, but the choice of basis is arbitrary. We can follow the convention of standard PCA by choosing an orthogonal matrix $O$ so that $(\widehat{V}O)^T S(\widehat{V}O)$ is diagonal, and take $\widehat{V}O$ as the orthonormal basis for the subspace estimate.

# 3 Theory

In this section we describe our theoretical framework for studying the statistical properties of $\widehat{X}$ given by (1) with *arbitrary* input matrices that satisfy the following assumptions.

**Assumption 1** (Symmetry). *$S$ and $\Sigma$ are $p \times p$ symmetric matrices.*

**Assumption 2** (Identifiability). *$\delta = \delta(\Sigma) = \lambda_d(\Sigma) - \lambda_{d+1}(\Sigma) > 0$.*

**Assumption 3** (Sparsity). *The projection $\Pi$ onto the subspace spanned by the eigenvectors of $\Sigma$ corresponding to its $d$ largest eigenvalues satisfies $\|\Pi\|_{2,0} \leq s$, or equivalently, $\|\operatorname{diag}(\Pi)\|_0 \leq s$.*

The key result (Theorem 3.1 below) implies that the statistical properties of the error of the estimator

$$\Delta := \widehat{X} - \Pi \,,$$

can be derived, in many cases, by routine analysis of the entrywise errors of the input matrix

$$W := S - \Sigma \,.$$

There are two main ideas in our analysis of $\widehat{X}$. The first is relating the difference in the values of the objective function in (1) at $\Pi$ and $\widehat{X}$ to $\Delta$. The second is exploiting the decomposability of the regularizer. Conceptually, this is the same approach taken by [25] in analyzing the statistical properties of regularized $M$-estimators. It is worth noting that the curvature result in our problem comes from the geometry of the constraint set in (1). It is different from the "restricted strong convexity" in [25], a notion of curvature tailored for regularization in the form of penalizing an unconstrained convex objective.

## 3.1 Variational analysis on the Fantope

The first step of our analysis is to establish a bound on the curvature of the objective function along the Fantope and away from the truth.

**Lemma 3.1** (Curvature). *Let $A$ be a symmetric matrix and $E$ be the projection onto the subspace spanned by the eigenvectors of $A$ corresponding to its $d$ largest eigenvalues $\lambda_1 \geq \lambda_2 \geq \cdots$. If $\delta_A = \lambda_d - \lambda_{d+1} > 0$, then*

$$\frac{\delta_A}{2} \|\!|E - F|\!\|_2^2 \leq \langle A, E - F \rangle$$

*for all $F$ satisfying $0 \preceq F \preceq I$ and $\operatorname{tr}(F) = d$.*

A version of Lemma 3.1 first appeared in [18] with the additional restriction that $F$ is a projection matrix. Our proof of the above extension is a minor modification of their proof.

The following is an immediate corollary of Lemma 3.1 and the Ky Fan maximal principle.

**Corollary 3.1** (A $\sin \Theta$ theorem [18]). *Let $A, B$ be symmetric matrices and $\mathcal{M}_A$, $\mathcal{M}_B$ be their respective $d$-dimensional principal subspaces. If $\delta_{A,B} = [\lambda_{d+1}(A) - \lambda_d(A)] \vee [\lambda_{d+1}(B) - \lambda_d(B)]$, then*

$$\|\!|\sin \Theta(\mathcal{M}_A, \mathcal{M}_B)|\!\|_2 \leq \frac{\sqrt{2}}{\delta_{A,B}} \|\!|A - B|\!\|_2 \,.$$

The advantage of Corollary 3.1 over the Davis-Kahan Theorem [see, e.g., 21, §VII.3] is that it does not require a bound on the differences between eigenvalues of $A$ and eigenvalues of $B$. This means that typical applications of the Davis-Kahan Theorem require the additional invocation of Weyl's Theorem. Our primary use of this result is to show that even if $\operatorname{rank}(\widehat{X}) \neq d$, its principal subspace will be close to that of $\Pi$ if $\Delta$ is small.

**Corollary 3.2** (Subspace error bound). *If $\mathcal{M}$ is the principal $d$-dimensional subspace of $\Sigma$ and $\widehat{\mathcal{M}}$ is the principal $d$-dimensional subspace of $\widehat{X}$, then*

$$\|\!|\sin \Theta(\mathcal{M}, \widehat{\mathcal{M}})|\!\|_2 \leq \sqrt{2} \|\!|\Delta|\!\|_2 \,.$$

## 3.2 Deterministic error

With Lemma 3.1, it is straightforward to prove the following theorem.

**Theorem 3.1** (Deterministic error bound). *If $\lambda \geq \|W\|_{\infty,\infty}$ and $s \geq \|\Pi\|_{2,0}$ then*
$$\|\!|\Delta|\!\|_2 \leq 4s\lambda/\delta \,.$$

Theorem 3.1 holds for any global optimizer $\widehat{X}$ of (1). It does not assume that the solution is rank-$d$ as in [11]. The next theorem gives a sufficient condition for support recovery by diagonal thresholding $\widehat{X}$.

**Theorem 3.2** (Support recovery). *For all $t > 0$*
$$\left|\{j : \Pi_{jj} = 0, \widehat{X}_{jj} \geq t\}\right| + \left|\{j : \Pi_{jj} \geq 2t, \widehat{X}_{jj} < t\}\right| \leq \frac{\|\!|\Delta|\!\|_2^2}{t^2} \,.$$

*As a consequence, the variable selection procedure $\widehat{J}(t) := \left\{j : \widehat{X}_{jj} \geq t\right\}$ succeeds if $\min_{j:\Pi_{jj}\neq 0} \Pi_{jj} \geq 2t > 2\|\!|\Delta|\!\|_2$.*

## 3.3 Statistical properties

In this section we use Theorem 3.1 to derive the statistical properties of $\widehat{X}$ in a generic setting where the entries of $W$ uniformly obey a restricted sub-Gaussian deviation inequality. This is not the most general result possible, but it allows us to illustrate the statistical properties of $\widehat{X}$ for two different types of input matrices: sample covariance and Kendall's tau correlation. The former is the standard input for PCA; the latter has recently been shown to be a useful robust and nonparametric tool for high-dimensional graphical models [26].

**Theorem 3.3** (General statistical error bound). *If there exists $\sigma > 0$ and $n > 0$ such that $\Sigma$ and $S$ satisfy*
$$\max_{ij} \mathbb{P}\left(|S_{ij} - \Sigma_{ij}| \geq t\right) \leq 2\exp\left(-4nt^2/\sigma^2\right) \tag{3}$$
*for all $t \leq \sigma$ and*
$$\lambda = \sigma\sqrt{\log p/n} \leq \sigma \,, \tag{4}$$
*then*
$$\|\!|\widehat{X} - \Pi|\!\|_2 \leq \frac{4\sigma}{\delta} s\sqrt{\log p/n}$$
*with probability at least $1 - 2/p^2$.*

**Sample covariance** Consider the setting where the input matrix is the sample covariance matrix of a random sample of size $n > 1$ from a sub-Gaussian distribution. A random vector $Y$ with $\Sigma = \mathrm{Var}(Y)$ has sub-Gaussian distribution if there exists a constant $L > 0$ such that
$$\mathbb{P}\left(|\langle Y - \mathbb{E}Y, u\rangle| \geq t\right) \leq \exp\left(-Lt^2/\|\Sigma^{1/2}u\|_2^2\right) \tag{5}$$
for all $u$ and $t \geq 0$. Under this condition we have the following corollary of Theorem 3.3.

**Corollary 3.3.** *Let $S$ be the sample covariance matrix of an i.i.d. sample of size $n > 1$ from a sub-Gaussian distribution (5) with population covariance matrix $\Sigma$. If $\lambda$ is chosen to satisfy (4) with $\sigma = c\lambda_1$, then*
$$\|\!|\widehat{X} - \Pi|\!\|_2 \leq C\frac{\lambda_1}{\delta} s\sqrt{\log p/n}$$
*with probablity at least $1 - 2/p^2$, where $c, C$ are constants depending only on $L$.*

Comparing with the minimax lower bounds derived in [17, 18], we see that the rate in Corollary 3.3 is roughly larger than the optimal minimax rate by a factor of
$$\sqrt{\lambda_1/\lambda_{d+1}} \cdot \sqrt{s/d}$$
The first term only becomes important in the near-degenerate case where $\lambda_{d+1} \ll \lambda_1$. It is possible with much more technical work to get sharp dependence on the eigenvalues, but we prefer to retain brevity and clarity in our proof of the version here. The second term is likely to be unimprovable without additional conditions on $S$ and $\Sigma$ such as a spiked covariance model. Very recently, [14] showed in a testing framework with similar assumptions as ours when $d = 1$ that the extra factor $\sqrt{s}$ is necessary for any polynomial time procedure if the planted clique problem cannot be solved in randomized polynomial time.

**Kendall's tau**  Kendall's tau correlation provides a robust and nonparametric alternative to ordinary (Pearson) correlation. Given an $n \times p$ matrix whose rows are i.i.d. $p$-variate random vectors, the theoretical and empirical versions of Kendall's tau correlation matrix are

$$\tau_{ij} := \mathrm{Cor}\left(\mathrm{sign}(Y_{1i} - Y_{2i}),\, \mathrm{sign}(Y_{1j} - Y_{2j})\right)$$

$$\hat{\tau}_{ij} := \frac{2}{n(n-1)} \sum_{s<t} \mathrm{sign}(Y_{si} - Y_{ti}) \, \mathrm{sign}(Y_{sj} - Y_{tj}).$$

A key feature of Kendall's tau is that it is invariant under strictly monotone transformations, i.e.

$$\mathrm{sign}(Y_{si} - Y_{ti}) \, \mathrm{sign}(Y_{sj} - Y_{tj})) = \mathrm{sign}(f_i(Y_{si}) - f_i(Y_{ti})) \, \mathrm{sign}(f_j(Y_{sj}) - f_j(Y_{tj})),$$

where $f_i, f_j$ are strictly monotone transformations. When $Y$ is multivariate Gaussian, there is also a one-to-one correspondence between $\tau_{ij}$ and $\rho_{ij} = \mathrm{Cor}(Y_{1i}, Y_{1j})$ [27] :

$$\tau_{ij} = \frac{2}{\pi} \arcsin(\rho_{ij}). \tag{6}$$

These two observations led [26] to propose using

$$\widehat{T}_{ij} = \begin{cases} \sin\left(\frac{\pi}{2}\hat{\tau}_{ij}\right) & \text{if } i \neq j \\ 1 & \text{if } i = j. \end{cases} \tag{7}$$

as an input matrix to Gaussian graphical model estimators in order to extend the applicability of those procedures to the wider class of *nonparanormal* distributions [28]. This same idea was extended to sparse PCA by [29]; they proposed and analyzed using $\widehat{T}$ as an input matrix to the non-convex sparse PCA procedure of [13]. A shortcoming of that approach is that their theoretical guarantees only hold for the global solution of an NP-hard optimization problem. The following corollary of Theorem 3.3 rectifies the situation by showing that $\widehat{X}$ with Kendall's tau is nearly optimal.

**Corollary 3.4.** *Let $S = \widehat{T}$ as defined in (7) for an i.i.d. sample of size $n > 1$ and let $\Sigma = T$ be the analogous quantity with $\tau_{ij}$ in place of $\hat{\tau}_{ij}$. If $\lambda$ is chosen to satisfy (4) with $\sigma = \sqrt{8}\pi$, then*

$$\|\widehat{X} - \Pi\|_2 \leq \frac{8\sqrt{2}\pi}{\delta} s\sqrt{\log p/n}$$

*with probablity at least $1 - 2/p^2$.*

Note that Corollary 3.4 only requires that $\hat{\tau}$ be computed from an i.i.d. sample. It does not specify the marginal distribution of the observations. So $\Sigma = T$ is not necessarily positive semidefinite and may be difficult to interpret. However, under additional conditions, the following lemma gives meaning to $T$ by extending (6) to a wide class of distributions, called *transelliptical* by [29], that includes the nonparanormal. See [29, 30] for further information.

**Lemma** ([29, 30]). *If $(Y_{11}, \ldots, Y_{1p})$ has continuous distribution and there exist monotone transformations $f_1, \ldots, f_p$ such that*

$$\left(f_1(Y_{11}), \ldots, f_p(Y_{1p})\right)$$

*has elliptical distribution with scatter matrix $\tilde{\Sigma}$, then*

$$T_{ij} = \tilde{\Sigma}_{ij} / \sqrt{\tilde{\Sigma}_{ii} \tilde{\Sigma}_{jj}}.$$

*Moreover, if $f_j(Y_{1j}), j = 1, \ldots, p$ have finite variance, then $T_{ij} = \mathrm{Cor}\left(f_i(Y_{1i}), f_j(Y_{1j})\right)$.*

This lemma together with Corollary 3.4 shows that Kendall's tau can be used in place of the sample correlation matrix for a wide class of distributions without much loss of efficiency.

## 4  An ADMM algorithm

The chief difficulty in directly solving (1) is the interaction between the penalty and the Fantope constraint. Without either of these features, the optimization problem would be much easier. ADMM can exploit this fact if we first rewrite (1) as the equivalent equality constrained problem

$$\begin{aligned} \text{minimize} \quad & \infty \cdot \mathbf{1}_{\mathcal{F}^d}(X) - \langle S, X \rangle + \lambda\|Y\|_{1,1} \\ \text{subject to} \quad & X - Y = 0, \end{aligned} \tag{8}$$

---
**Algorithm 1** Fantope Projection and Selection (FPS)
---
**Require:** $S = S^T, d \geq 1, \lambda \geq 0, \rho > 0, \epsilon > 0$
   $Y^{(0)} \leftarrow 0, U^{(0)} \leftarrow 0$                                        ▷ Initialization
   **repeat**   $t = 0, 1, 2, 3, \ldots$
       $X^{(t+1)} \leftarrow \mathcal{P}_{\mathcal{F}^d}\big(Y^{(t)} - U^{(t)} + S/\rho\big)$                   ▷ Fantope projection
       $Y^{(t+1)} \leftarrow \mathcal{S}_{\lambda/\rho}\big(X^{(t+1)} + U^{(t)}\big)$            ▷ Elementwise soft thresholding
       $U^{(t+1)} \leftarrow U^{(t)} + X^{(t+1)} - Y^{(t+1)}$                ▷ Dual variable update
   **until** $\max(\|\|X^{(t)} - Y^{(t)}\|\|_2^2 \,,\, \rho^2 \|\|Y^{(t)} - Y^{(t-1)}\|\|_2^2) \leq d\epsilon^2$     ▷ Stopping criterion
   **return** $Y^{(t)}$
---

in the variables $X$ and $Y$, where $\mathbf{1}_{\mathcal{F}^d}$ is the 0-1 indicator function for $\mathcal{F}^d$ and we adopt the convention $\infty \cdot 0 = 0$. The augmented Lagrangian associated with (8) has the form

$$\mathcal{L}_\rho(X, Y, U) := \infty \cdot \mathbf{1}_{\mathcal{F}^d}(X) - \langle S, X \rangle + \lambda \|Y\|_{1,1} + \frac{\rho}{2}\Big(\|\|X - Y + U\|\|_2^2 - \|\|U\|\|_2^2\Big), \quad (9)$$

where $U = (1/\rho)Z$ is the scaled ADMM dual variable and $\rho$ is the ADMM penalty parameter [see 19, §3.1]. ADMM consists of iteratively minimizing $\mathcal{L}_\rho$ with respect to $X$, minimizing $\mathcal{L}_\rho$ with respect to $Y$, and then updating the dual variable. Algorithm 1 summarizes the main steps.

In light of the separation of $X$ and $Y$ in (9) and some algebraic manipulation, the $X$ and $Y$ updates reduce to computing the proximal operators

$$\mathcal{P}_{\mathcal{F}^d}\big(Y - U + S/\rho\big) := \underset{X \in \mathcal{F}^d}{\arg\min} \frac{1}{2}\|\|X - (Y - U + S/\rho)\|\|_2^2$$

$$\mathcal{S}_{\lambda/\rho}(X + U) := \underset{Y}{\arg\min} \frac{\lambda}{\rho}\|Y\|_{1,1} + \frac{1}{2}\|\|(X + U) - Y\|\|_2^2.$$

$\mathcal{S}_{\lambda/\rho}$ is the elementwise soft thresholding operator [e.g., 19, §4.4.3] defined as

$$\mathcal{S}_{\lambda/\rho}(x) = \text{sign}(x) \max(|x| - \lambda/\rho, 0).$$

$\mathcal{P}_{\mathcal{F}^d}$ is the Euclidean projection onto $\mathcal{F}^d$ and is given in closed form in the following lemma.

**Lemma 4.1** (Fantope projection). *If $X = \sum_i \gamma_i u_i u_i^T$ is a spectral decomposition of $X$, then $\mathcal{P}_{\mathcal{F}^d}(X) = \sum_i \gamma_i^+(\theta) u_i u_i^T$, where $\gamma_i^+(\theta) = \min(\max(\gamma_i - \theta, 0), 1)$ and $\theta$ satisfies the equation $\sum_i \gamma_i^+(\theta) = d$.*

Thus, $\mathcal{P}_{\mathcal{F}^d}(X)$ involves computing an eigendecomposition of $Y$, and then modifying the eigenvalues by solving a monotone, piecewise linear equation.

Rather than fix the ADMM penalty parameter $\rho$ in Algorithm 1 at some constant value, we recommend using the varying penalty scheme described in [19, §3.4.1] that dynamically updates $\rho$ after each iteration of the ADMM to keep the primal and dual residual norms (the two sum of squares in the stopping criterion of Algorithm 1) within a constant factor of each other. This eliminates an additional tuning parameter, and in our experience, yields faster convergence.

## 5 Simulation results

We conducted a simulation study to compare the effectiveness of FPS against three deflation-based methods: DSPCA (which is just FPS with $d = 1$), GPower$_{\ell_1}$ [7], and SPC [5, 6]. These methods obtain multiple component estimates by taking the $k$th component estimate $\hat{v}_k$ from input matrix $S_k$, and then re-running the method with the deflated input matrix: $S_{k+1} = (I - \hat{v}_k \hat{v}_k^T) S_k (I - \hat{v}_k \hat{v}_k^T)$. The resulting $d$-dimensional principal subspace estimate is the span of $\hat{v}_1, \ldots, \hat{v}_d$. Tuning parameter selection can be much more complicated for these iterative deflation methods. In our simulations, we simply fixed the regularization parameter to be the same for all $d$ components.

We generated input matrices by sampling $n = 100$, i.i.d. observations from a $\mathcal{N}_p(0, \Sigma), p = 200$ distribution and taking $S$ to be the usual sample covariance matrix. We considered two different types of sparse $\Pi = VV^T$ of rank $d = 5$: those with *disjoint* support for the nonzero entries of the

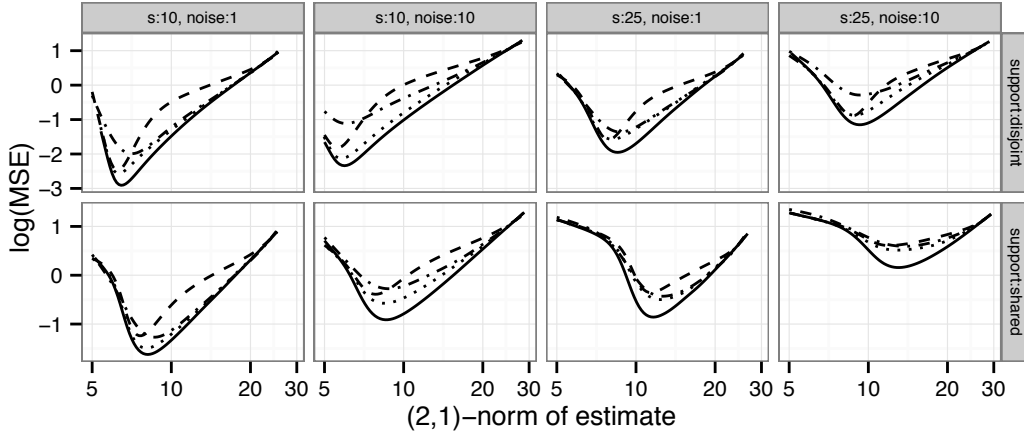

Figure 1: Mean squared error of FPS (———), DSPCA with deflation ( − − ), GPower$_{\ell_1}$ (·······), and SPC (·−·−) across 100 replicates each of a variety of simulation designs with $n = 100$, $p = 200$, $d = 5$, $s \in \{10, 25\}$, noise $\sigma^2 \in \{1, 10\}$.

columns of $V$ and those with *shared* support. We generated $V$ by sampling its nonzero entries from a standard Gaussian distribution and then orthnormalizing $V$ while retaining the desired sparsity pattern. In both cases, the number of nonzero rows of $V$ is equal to $s \in \{10, 25\}$. We then embedded $\Pi$ inside the population covariance matrix $\Sigma = \alpha \Pi + (I - \Pi)\Sigma_0(I - \Pi)$, where $\Sigma_0$ is a Wishart matrix with $p$ degrees of freedom and $\alpha > 0$ is chosen so that the effective noise level (in the optimal minimax rate [18]), $\sigma^2 = \sqrt{\lambda_1 \lambda_{d+1}}/(\lambda_d - \lambda_{d+1}) \in \{1, 10\}$.

Figure 1 summarizes the resulting mean squared error $\|\|\widehat{\Pi} - \Pi\|\|_2^2$ across 100 replicates for each of the different combinations of simulation parameters. Each method's regularization parameter varies over a range and the $x$-axis shows the $(2, 1)$-norm of the corresponding estimate. At the right extreme, all methods essentially correspond to standard PCA. It is clear that regularization is beneficial, because all the methods have significantly smaller MSE than standard PCA when they are sufficiently sparse. Comparing between methods, we see that FPS dominates in all cases, but the competition is much closer in the disjoint support case. Finally, all methods degrade when the number of active variables or noise level increases.

# 6  Discussion

Estimating sparse principal subspaces in high-dimensions poses both computational and statistical challenges. The contribution of this paper—a novel SDP based estimator, an efficient algorithm, and strong statistical guarantees for a wide array of input matrices—is a significant leap forward on both fronts. Yet, there are newly open problems and many possible extensions related to this work. For instance, it would be interesting to investigate the performance of FPS a under weak, rather than exact, sparsity assumption on $\Pi$ (e.g., $\ell_q$, $0 < q \leq$ sparsity). The optimization problem (1) and ADMM algorithm can easily be modified handle other types of penalties. In some cases, extensions of Theorem 3.1 would require minimal modifications to its proof. Finally, the choices of dimension $d$ and regularization parameter $\lambda$ are of great practical interest. Techniques like cross-validation need to be carefully formulated and studied in the context of principal subspace estimation.

**Acknowledgments**

This research was supported in part by NSF grants DMS-0903120, DMS-1309998, BCS-0941518, and NIH grant MH057881.

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
