[Supplementary Material]

# A   Appendix - Supplemental Material

*Proof of Lemma 2.1.*  By the Cauchy-Schwarz Inequality,

$$\|VV^T\|_{1,1} = \sum_{ijk}|V_{ik}||V_{jk}| \leq \sum_{ij}\left[\sum_k|V_{ik}|^2\right]^{1/2}\left[\sum_k|V_{jk}|^2\right]^{1/2} = \|V\|_{2,1}^2\,.$$

Since the norms of the row of $V/\|V\|_{2,2}$ are $\leq 1$,

$$\|V\|_{2,1} = \left\|V/\|V\|_{2,2}\right\|_{2,1}\|V\|_{2,2} \leq \|V\|_{2,0}\|V\|_{2,2}\,.$$

Using the identity $\|V\|_{2,2}^2 = \mathrm{tr}(VV^T)$ and the preceding inequalities,

$$\|VV^T\|_{1,1} \leq \|V\|_{2,1}^2 \leq \|V\|_{2,0}^2\|V\|_{2,2}^2 = \|V\|_{2,0}^2\,\mathrm{tr}(VV^T)\,. \qquad \blacksquare$$

*Proof of Lemma 3.1.*  We first assume that $A \succeq 0$. Using the spectral decomposition of $A$ and the assumptions that $0 \preceq F \preceq I$ and $\mathrm{tr}(F) \leq d$, it is straightforward to show that

$$\begin{aligned}
\langle A, E - F\rangle &= \langle EA, I - F\rangle - \langle (I-E)A, F\rangle \\
&\geq \lambda_d\langle E, I - F\rangle - \lambda_{d+1}\langle I - E, F\rangle \\
&= \delta(d - \langle E, F\rangle)\,.
\end{aligned}$$

Now $0 \preceq E \preceq I$ and $0 \preceq F \preceq I$. So

$$\begin{aligned}
2(d - \langle E, F\rangle) &= \mathrm{tr}(E) + \mathrm{tr}(F) - 2\langle E, F\rangle \\
&\geq \|\!|E|\!\|_2^2 + \|\!|F|\!\|_2^2 - 2\langle E, F\rangle \\
&= \|\!|E - F|\!\|_2^2\,.
\end{aligned}$$

If $A$ is not positive semidefinite, then we may choose $c > 0$ sufficiently large so that $A + cI \succeq 0$. Note that $A + cI$ has the same spectral gap as $A$ and $\langle A + cI, E - F\rangle = \langle A, E - F\rangle$. So the indefinite case follows from the positive semidefinite case. $\qquad \blacksquare$

*Proof of Corollary 3.2.*  The definition of the Fantope ensures that $\mathrm{rank}(\widehat{X}) \geq d$, so $\widehat{X}$ does have a principal $d$-dimensional subspace (though not necessarily unique). Since $\Pi$ is a rank-$d$ projection matrix, $\lambda_d(\Pi) - \lambda_{d+1}(\Pi) = 1$. Now apply Corollary 3.1. $\qquad \blacksquare$

*Proof of Theorem 3.1.*  Since $\widehat{X}$ is optimal and $\Pi$ is feasible for (1),

$$0 \leq \langle S, \Delta\rangle - \lambda(\|\Pi + \Delta\|_{1,1} - \|\Pi\|_{1,1})\,.$$

On the otherhand, Lemma 3.1 implies

$$\frac{\delta}{2}\|\!|\Delta|\!\|_2^2 \leq -\langle \Sigma, \Delta\rangle\,.$$

Thus,

$$\begin{aligned}
\frac{\delta}{2}\|\!|\Delta|\!\|_2^2 &\leq \langle W, \Delta\rangle - \lambda(\|\Pi + \Delta\|_{1,1} - \|\Pi\|_{1,1}) \\
&\leq \|W\|_{\infty,\infty}\|\Delta\|_{1,1} - \lambda(\|\Pi + \Delta\|_{1,1} - \|\Pi\|_{1,1}) \\
&\leq \lambda(\|\Delta\|_{1,1} - \|\Pi + \Delta\|_{1,1} + \|\Pi\|_{1,1})\,.
\end{aligned}$$

Let $J$ be the subset of indices of the nonzero entries of $\Pi$. For a symmetric matrix $B$, we write $B_J$ for the matrix equal to $B$ on $J$ and zero off of $J$. Then $\|B\|_{1,1} = \|B_J\|_{1,1} + \|B - B_J\|_{1,1}$ and $\Pi = \Pi_J$. So

$$\begin{aligned}
\|\Delta\|_{1,1} - \|\Pi + \Delta\|_{1,1} + \|\Pi\|_{1,1} &= \|\Delta_J\|_{1,1} - \|\Pi_J + \Delta_J\|_{1,1} + \|\Pi_J\|_{1,1} \\
&\leq 2\|\Delta_J\|_{1,1}\,,
\end{aligned}$$

where the second line is the triangle inequality. Since $\Delta_J$ has at most $s^2$ nonzero entries,

$$\|\Delta_J\|_{1,1} \leq s\|\!|\Delta_J|\!\|_2 \leq s\|\!|\Delta|\!\|_2\,. \qquad \blacksquare$$

*Proof of Theorem 3.2.* Clearly,

$$D_0 := \{j : \Pi_{jj} = 0, \widehat{X}_{jj} \geq t\} \subseteq \{j : |\Delta_{jj}| \geq t\},$$
$$D_1 := \{j : \Pi_{jj} \geq 2t, \widehat{X}_{jj} < t\} \subseteq \{j : |\Delta_{jj}| \geq t\},$$

and $D_0 \cap D_1 = \emptyset$. Then by Markov's Inequality,

$$|D_0| + |D_1| \leq \left|\{j : |\Delta_{jj}| \geq t\}\right| \leq \frac{1}{t^2}\sum_j |\Delta_{jj}|^2 \leq \frac{\|\!|\Delta\|\!|_2^2}{t^2}\,. \qquad \blacksquare$$

*Proof of Theorem 3.3.* We have by (3) and the union bound that

$$\mathbb{P}\left(\|W\|_{\infty,\infty} \geq \lambda\right) \leq 2\exp\left(-4\log p + 2\log p\right) = 2/p^2\,,$$

and Theorem 3.1 yields the desired result. $\qquad \blacksquare$

*Proof of Corollary 3.3.* Note that $\|\Sigma^{1/2}u\|_2^2 \leq \lambda_1\|u\|_2^2$. Under assumption (5), it can be shown by Bernstein's Inequality [see 1, Lemma 2.2.11] that $S - \Sigma$ satisfies (3) with $\sigma = c\lambda_1$ where $c > 0$ is a constant depending only on $L$. The assumption that $\log p \leq n$ in (4) ensures that only the moderate sub-Gaussian deviation in Bernstein's Inequality is active. $\qquad \blacksquare$

*Proof of Corollary 3.4.* Liu et al. [2, Theorem 4.2] use Hoeffding's Inequality for U-statistics to show that

$$\max_{ij}\mathbb{P}\left(|S_{ij} - \Sigma_{ij}| > t\right) \leq 2\exp\left(-4nt^2/\sigma^2\right)\,. \qquad \blacksquare$$

*Proof of Lemma 4.1.* Let $V$ denote the matrix whose columns are the eigenvectors of $X$. Since the Frobenius norm and Fantope are orthogonally invariant,

$$\mathcal{P}_{\mathcal{F}^d}(X) = \arg\min_{Y \in \mathcal{F}^d} \frac{1}{2}\|\!|X - Y\|\!|_2^2 = V\left[\arg\min_{0 \preceq y \preceq 1, \langle y, \mathbf{1}\rangle = d}\frac{1}{2}\|\!|\gamma - y\|\!|_2^2\right]V^T\,.$$

The Lagrangian associated with the problem above is

$$\frac{1}{2}\|\!|\gamma - y\|\!|_2^2 + \langle y - \mathbf{1}, \tau_1\rangle - \langle y, \tau_0\rangle + \theta\left(\langle y, \mathbf{1}\rangle - d\right)\,,$$

which upon differentiation with respect to $y$ and comparing to 0 yields the optimality condition

$$y - \gamma + \tau_1 - \tau_0 + \theta\mathbf{1} = 0\,.$$

By complementary slackness, if $0 < y_i < 1$ then $\tau_{0i} = \tau_{1i} = 0$ and $y_i = \gamma_i - \theta$. Thus, the optimal value of $y$ must satisfy

$$\sum_i \min(\max(y_i - \theta, 0), 1) = d\,. \qquad \blacksquare$$

**Additional references**

[1]    A. W. van der Vaart and J. A. Wellner. *Weak convergence and empirical processes.* Springer-Verlag, 1996.
[2]    H. Liu et al. "High-dimensional semiparametric gaussian copula graphical models ". In: *Ann. Statis.* 40.4 (2012), pp. 2293–2326.