[Reviews · NeurIPS 2013]

Submitted by Assigned_Reviewer_7

The authors address the problem of sparse principal component analysis when more than a single component is to be estimated. The standard method computes the principal components one-by-one and uses a heuristic step often called "deflation" to switch to the next component. This step-by-step sequential method is fragile and criticizing it is absolutely natural. The authors suggest to estimate the matrix using projection to the Fantope: the convex relaxation of the orthogonal matrices.

The authors provide an efficient algorithmic scheme to solve the problem and analyse the solution statistically. The paper is well written and will interest people who are working on this very popular topic.

An important question: why is the orthogonality constraint crucial in sparse PCA? in fact in standard (desne / full) PCA it is a consequence of the fact that the low-rank approximation of a matrix is provided by the eigenvalue decomposition which has orthogonal factors. Somehow when an extra sparsity assumption is maid why should we keep the orthogonality?

The experimental section is not convincing. I would have wished to see a phase-transition type of diagram to see when the performance really outperforms the rival as the factors supports overlap. The idea of computing multiple principal components at once had already been tackled by Journee et al. (ref [8] in the paper, please name all the authors by the way in [8]). Why is there no comparison with their method? the code for their method is available online. There should be a numerical comparison with a sparse matrix factorization method as well. Comparison of the supports found by each method on real data may be interesting.


Summary: The paper is about a relevant problem, is well written and suggests a method which seems to be efficient for the goal fixed by the authors. I would like however a more convincing discussion on the orthogonality constraints which seem more embarrassing than useful. The algorithm is an incremental update of DSPCA [1] using Fantope constraint, and theoretical results are also mostly incremental, and not so exciting.

Submitted by Assigned_Reviewer_8

The authors consider the Sparse PCA problem, i.e., PCA with the assumption that the "principal vectors" depend on only a few of the variables. They propose a convex relaxation for solving this problem and analyze its performance. An ADMM based method is also developed for solving the resulting SDP efficiently. Finally they show theoretical results when a Kendall's tau matrix is used as input instead of the usual sample covariance matrix.

The paper is clearly written and the authors have done a good job in explaining various concepts as they become used in their exposition. The results presented in this paper appear to be correct.

I like the convex formulation. The techniques used to achieve it are simple and this might certainly be considered one of the strengths of the paper. It also goes along nicely with the ADMM formulation and Lemma 4.1. I have a minor suggestion : it would be nice if the authors mention the fact that they have a closed form expression for the projection onto the fantope a little earlier in the paper (possibly along with the motivation of a convex relaxation).

However, the statistical analysis seems a little weak, particularly in the near low rank and low rank cases. If there were only d non zero eigenvalues, the bound the authors present could potentially be very far away from the minimax bound. On a related note, the authors should consider rephrasing the statement "It is possible (with more technical work) to tighten the bound..." in the last paragraph of page 5; it does not appear to contribute constructively without a discussion about what this technical work could be.

Even if the Appendix contains all the proofs, it would be helpful if the authors present the reader with a small proof sketch after each result has been stated (also note that reviewers are not required to read the supplementary material).

The section on Simulation Results again seems a little weak. The figures need reworking as Figure 1(a) is just too hard to see on a printed version of the paper. On a related note, it will be nice to explain (in the text and/or in the caption) what "overlapping" and "non-overlapping" sparsity patterns mean precisely. It will also be helpful to insert an intuitive explanation as to why the performance is so different in these cases and when it matters.

Minor point : The authors should consider rephrasing ".. has a wide range of applications - Science, engineering, ... ". It would be much more helpful to point the reader to a few specific applications with references.
Summary: The authors propose a simple convex relaxation (with an efficient solution method) for the Sparse PCA problem. The paper is written well but the results could definitely use some intuitive justification/proof sketch in the main text. The convex relaxation (and the attendant ADMM algorithm) is nice. On the other hand, while the the performance of their algorithm is shown to be near optimal, it is not satisfactory in some natural cases like the low rank or the near low rank settings and the simulation results do not seem extremely persuasive.

Submitted by Assigned_Reviewer_9

The paper introduced a novel formulation of sparse subspace discovery problem as one of finding sparse matrices in a fanotope, a convex hull of rank-d projection matrices. The proposed formulation gives rise to a convex problem that is solved using an ADMM algorithm. Guarantees for support recovery and error in frobenius norm are provided. Finally, a set of illustrative synthetic experiments are provided demonstrating improved performance, in terms of frobenius norm, in recovering a sparse factorization of a target matrix.
The new formulation is elegant and provides a novel and intuitive replacement for the sparse PCA objective. The drop-in replacements for a covariance matrix, Kendal and Pearson correlation, can also be nearly-optimally decomposed enabling an analog of non-linear sparse PCA with the usual constraints familiar from the nonparanormal work.
A bit lengthier discussion of the synthetic experiments would be helpful. The numbers of selected variables are quite high compared to what they should be for matrix Pi, which seems to be of rank 5 and fairly sparse itself. So, what is the optimal sparse decomposition for the overlap and non-overlap examples and does the method achieve this decomposition with 100s of selected variables? How well does DSPCA work in this respect? The Frobenius norm as an error term is just a part of the story. The claim that FPS gives rise to estimates that “are typically sparser” is not supported by the results. Please provide more information here.
Notation: Please define what \vee stands for. The meaning can be gleaned from the context, but it is sufficiently non-standard that it would benefit from a clear definition.
Couple of minor comments: “difficult to interepretation” -> “difficult to interpret”, Figure 1 use the same notation for Frobenius norm as in the rest of the paper
Summary: Clearly written paper with novel contribution in recasting sparse PCA problem and providing a straightforward new method for estimating the sparse subspaces.
Author Feedback

Author rebuttal: We thank the referees for their comments on our paper. We had three main aims in writing this paper:
(1) naturally extend DSPCA to estimate multi-dimensional subspaces with convex optimization.
(2) unify statistical theory for both DSPCA and our extension.
(3) efficiently solve the convex optimization problem (with ADMM); there is essentially no additional cost compared to the d=1 case of DSPCA.

The bulk of our paper is dedicated to a unified theoretical analysis that provides the following novel contributions:
(i) Statistical guarantees for a computationally tractable estimator. Previous papers give guarantees for estimators that require solving an NP-hard optimization problem.
(ii) First rigorous statistical convergence rates for DSPCA (and our extension to subspaces) under a general statistical model and without controversial rank-1 conditions on the solution.
(iii) The theory extends beyond covariance matrices.

Our theoretical treatment provides the first rigorous and general statistical error rates for sparse PCA and subspace estimators using convex relaxation. In particular, our theory resolves the controversial rank-1 condition of Amini & Wainwright (AoS 2009); their statistical estimation results only apply when the DSPCA solution is exactly rank one. Recent work by Krauthgamer, Nadler and Vilenchik (arXiv:1306.3690, June 13) has shown that in general, the DSPCA solution is rank one with very small probability, rendering the results in AW09 of questionable relevance. Our Theorem 3.3 shows that DSPCA is near-optimal, regardless of the solution’s rank.

The generality of Theorem 3.3 is significant because it has applications beyond sample covariance matrices. Recent extensions of Sparse PCA to non- and semi-parametric measures of correlation (e.g. Han & Liu, 2012) are based on NP-hard optimization problems. The theoretical guarantees in those works are conditional on being able to solve NP-hard problems. Theorem 3.3 and Corollary 3.4 shows that convex relaxation can be applied to those situations with provable guarantees.

The reviewers brought up the following major points:

1) Limitations of simulation results (all reviewers)
2) Necessity of orthogonality constraints (Assigned_Reviewer_7):
3) Missing intuition and proof sketches (Assigned_Reviewer_8)
4) Performance in the exact or near low rank case (Assigned_Reviewer_8)

We address these below.

* Simulation results (all reviewers)

As mentioned above, our paper focused especially on theoretical developments. So the section on simulation results was necessarily abbreviated by space limitations. The main point of the simulation results is to compare the efficiency of estimating a subspace directly (our method) with the more popular deflation approach based on DSPCA. The results show that there are significant gains in statistical efficiency. (Note that the y-axis of Figure 1B is on a logarithmic scale.) Moreover, the computational benefit of our approach over deflation is that there is essentially no additional cost over the d = 1 case, whereas deflation requires d-times the computation.

* Orthogonality (Assigned_Reviewer_7)

We agree that the relevance of orthogonality for sparse PCA is unclear, and we do not advocate orthogonality constraints. Instead, we sidestep the issue by using the Fantope to directly estimate the subspace. Because we only estimate the subspace (through its projection matrix), we are agnostic to any specific basis and thus do not make any restrictions related to orthogonality.



* Intuition and proof sketches (Assigned_Reviewer_8)

The theoretical development makes use of a curvature lemma (3.1). The proof relies on elementary linear algebra, but is both novel and powerful. The main theorem (3.1) is a combination of this curvature lemma and the standard Lasso argument (e.g. Negahban et al. 2012). This explanation appears in the introductory paragraph of Section 3. Our other results have very short proofs, but because of space constraints we had to put them in the supplement.


* Exact or near low rank case (Assigned_Reviewer_8)

The referee is correct to point out that our result does not achieve the minimax rate in terms of \lambda_{d+1}. However, if \lambda_{d+1} is very small, then estimation is already relatively easy with standard PCA. In fact, if the population covariance is exactly low rank, then there is no need for regularization and the problem is effectively low-dimensional.

The exact optimal rate has not been attained by any computationally tractable estimators in the current literature. This is potentially due to the fact that the “computational lower bound” (what can be obtained in polynomial time, see Berthet & Rigollet, 2013) is larger than the minimax lower bound.